# Lack of Awareness of Own Hypercholesterolemia or Statin Medication among Adult Statin Users in the United States: Prevalence and Patient Characteristics in a Repeated Cross-Sectional Study

**DOI:** 10.3390/ijerph19106099

**Published:** 2022-05-17

**Authors:** Kenjiro Imai, Takehiro Sugiyama, Mitsuru Ohsugi, Masafumi Kakei, Kazuo Hara

**Affiliations:** 1Diabetes and Metabolism Information Center, Research Institute, National Center for Global Health and Medicine, Tokyo 162-8655, Japan; keimai@hosp.ncgm.go.jp (K.I.); tsugiyama@hosp.ncgm.go.jp (T.S.); moosugi@hosp.ncgm.go.jp (M.O.); 2Division of General Medicine, Graduate School of Medicine, Jichi Medical University, Tochigi 329-0498, Japan; 3Department of Health Services Research, Faculty of Medicine, University of Tsukuba, Ibaraki 305-8577, Japan; 4Institute for Global Health Policy Research, Bureau of International Health Cooperation, National Center for Global Health and Medicine, Tokyo 162-8655, Japan; 5Department of Diabetes, Endocrinology and Metabolism, Center Hospital, National Center for Global Health and Medicine, Tokyo 162-8655, Japan; 6Minamiuonuma City Hospital, Niigata 949-6680, Japan; mmkakei@gmail.com; 7Division of Endocrinology and Metabolism, Department of Comprehensive Medicine 1, Saitama Medical Center, Jichi Medical University, Saitama 330-0834, Japan

**Keywords:** hypercholesterolemia, statin, prescription medication, lack of awareness

## Abstract

Knowledge of a patient’s medication is important in treating hyperlipidemia; however, little is known about this in practice. We carried out a repeated cross-sectional study to analyze a nationally representative sample of US adult statin users from the National Health and Nutrition Examination Survey, 1999–2018. We used medication bottle checks and self-reported survey data to estimate the percentage of individuals who are unaware of their hypercholesterolemia, type of medication, or how to take their medication. We used logistic regression to examine their characteristics. We included 8798 statin users; however, 17.6% were unaware of their hypercholesterolemia or statin use. Being older, male, non-Hispanic Black, taking a wider range of prescription medications, and previous diabetes or cardiovascular disease diagnosis were associated with lack of awareness. Serum low-density lipoprotein cholesterol level was lower among those lacking awareness (85.5 vs. 100.7 mg/dL; *p* < 0.001). Many of those unaware of drug type had been given little information about statins; 34.0% had no diagnosis of diabetes or cardiovascular disease, and of these, 27.1% were >75 years old. Roughly one in six lacked awareness, but no association was found with hypercholesterolemia control. Healthcare providers should ascertain a patient’s understanding and consider the risks and benefits of statin medication.

## 1. Introduction

An extremely high number of people in the United States are prescribed statins. In 2009–2010, 16.5% of the non-pregnant population aged ≥20 years were found to have been statin users [1], and 27.8% of the population aged ≥40 years (39.2 million people) were found to have used statins in 2012–2013 [2]. Many studies have demonstrated the beneficial effects of statins in lowering low-density lipoprotein cholesterol (LDL-C) levels [3,4], and statins have been shown to be more effective than other pharmacotherapies [5]. Conventionally, hypercholesterolemia has been treated with the “treat-to-target approach”, which sets a target LDL-C value in accordance with the degree of risk [6]. In 2002, the “fire-and-forget approach” of administering strong statins without setting a target LDL-C value was proposed for patients at risk of arteriosclerotic cardiovascular disease (ASCVD) [6]. The treat-to-target approach, however, remained recommended in guidelines until the American College of Cardiology/American Heart Association guidelines on the treatment of blood cholesterol were released in 2013 [7]. The guideline does not recommend using target LDL-C values in treatment; instead, it recommends statin treatment initiation values. Notably, statin use for ASCVD and diabetes is recommended to begin at the time of diagnosis. This strategy is maintained in the 2019 guidelines [8].

A study conducted in Lausanne, Switzerland, offered a different perspective, reporting that people with dyslipidemia on a self-reported hypolipidemic diet had a healthier dietary intake compared with the general population [9]. Recognizing one’s own risk of lifestyle-related diseases, thus, may be an important aspect of treatment strategies.

Anecdotal evidence from clinical practice suggests that, despite being treated with statins, a certain number of patients remain unaware of their hypercholesterolemia for which the statin medication has been prescribed. Among 141 patients awaiting carotid endarterectomy, 22% had either never had their lipid profiles checked or were unaware of having such tests (although all 37 patients on statins in this small-scale study were aware of their hypercholesterolemia) [10]. However, to our knowledge, there are no reports based on nationally representative data that focus on patients who lack awareness concerning their hypercholesterolemia or statin medication. For example, it is unknown whether a lack of awareness is negatively associated with the patient’s health condition.

In this context, using data from the National Health and Nutrition Examination Survey (NHANES), we aimed to estimate the lack of awareness of one’s own hypercholesterolemia status or statin medication among US adult statin users and to examine the characteristics of these patients. This covered the period of 1999–2018, during which the fire-and-forget approach was proposed (in 2002) and later recommended in the guidelines (in 2013).

## 2. Materials and Methods

### 2.1. Data Sources and the Study Population

We conducted a repeated cross-sectional study using data from the 1999–2018 NHANES [11]. The US National Center for Health Statistics, part of the Centers for Disease Control and Prevention, conducted the NHANES. Written informed consent was obtained from all participants, and the National Center for Health Statistics Research Ethics Review Board approved the NHANES protocols [12]. The NHANES used a stratified, multistage probability sampling design that enabled samples to represent the US civilian and non-institutional population. Data were collected at the respondent’s home or at mobile examination centers [13].

Among adults in the NHANES from 1999 to 2018, the overall unweighted response rate (calculated by the authors from response rates by survey year) [14] was 68.5% for household interviews and 65.1% at the mobile examination centers. The present study included data from individuals ≥20 years old. We excluded pregnant women from our analyses because statins are contraindicated during pregnancy. We also excluded non-statin users (n = 35,678). Irrespective of blood cholesterol level findings, statin therapy has been recommended for individuals ≤75 years old and who have clinical ASCVD [7]. Because these patients were recommended to take statins regardless of awareness of their statin use, we excluded those who responded “no”, “don’t know”, or “refused” to the question of whether they had ever had their blood cholesterol checked (n = 199) (Figure 1).

### 2.2. Statin Use

Interviewers in households asked survey participants if they had taken prescription medications in the preceding month. If the answer was “yes,” the participants were asked to show the interviewer all their medication containers. Medication names were entered and automatically matched with a prescription drug database (Lexicon Plus). Participants who did not have containers were asked to verbally list their medications. The interviewers’ original entries and matched database drug names were saved as separate variables for quality control purposes. We identified eight types of statin ingredients prescribed for NHANES respondents: lovastatin, simvastatin, pravastatin, fluvastatin, atorvastatin, cerivastatin, rosuvastatin, and pitavastatin. Statin use was defined by whether the statin came from a specific pill or a fixed-dose combination. We then divided participants into statin users and nonusers.

### 2.3. Lack of Patient Awareness about Their Hypercholesterolemia Status or Statin Use

Participants were asked if they had ever had their blood cholesterol checked. Those responding “yes” were asked whether a doctor or other health professional had ever told them their blood cholesterol level was high. Those responding “yes” to this question were then asked whether a doctor or other health professional had instructed them to take prescribed medicines to lower their cholesterol. Those again responding “yes” were asked whether they were taking prescribed medicine based on this advice [15]. A lack of awareness of statin users’ own hypercholesterolemia status was determined when they responded “no” or “don’t know” to the question of whether a doctor or other health professional had ever told them their blood cholesterol level was high (n = 1517). Statin users who responded “no” or “don’t know” to the question of whether a doctor or other health professional had instructed them to take prescribed medicines to lower their cholesterol levels (n = 165) were determined to have a lack of awareness regarding directions for taking prescribed medicines. Statin users who had been advised of their hypercholesterolemia level and responded “no” to the question of whether they were taking prescribed medicine for their condition (n = 103) were determined as having a lack of awareness regarding their medication. We regarded a lack of awareness about hypercholesterolemia or statin use as indicating a lack of awareness of one’s own hypercholesterolemia status, directions for taking prescribed medicines, or specifics of medication (n = 1785) (Figure 1).

### 2.4. Covariates

We extracted data on the following covariates: age, sex, race and ethnicity, language usually spoken, educational attainment, total cholesterol level, LDL-C level, body mass index (calculated as weight in kilograms divided by height in meters squared), number of people per respondent household, economic status, health insurance, routine place to go for healthcare, number of prescription drugs taken, caloric intake, fat intake, smoking status, and a diagnosis of hypertension, diabetes, cardiovascular disease, and/or stroke. We classified subjects into four age groups: 20–40, 41–60, 61–75, and >75 years old. Race and ethnicity were classified as non-Hispanic White, non-Hispanic Black, Mexican American, and other, including other Hispanics and multiracial participants. Language usually spoken was classified as English, Spanish, or other. We categorized educational attainment into above high school graduate, high school graduate or GED certificate, and less than high school graduate. LDL-C levels were calculated using the Friedewald equation [16] (total cholesterol—high-density lipoprotein cholesterol—triglycerides/5) for participants examined in the morning in a fasting state and showing triglyceride levels of ≤400 mg/dL. Each mobile examination participant was randomly selected to attend a morning, afternoon, or evening session at which height and weight were measured. We divided the number of people per household into three groups: 1, 2, and ≥3 people. Economic status was determined by a ratio of family income to the poverty line (income-to-poverty ratio (IPR)), which was divided into four categories: ≥4, 2 to <4, 1 to <2, and <1.

Routine place to go for healthcare was classified as clinic or health center, doctor’s office, or health maintenance organization, other place including hospital emergency rooms, and nowhere. We grouped the number of prescription drugs into three groups: 1, 2–4, and ≥5. Caloric and fat intake were ascertained by trained interviewers who conducted a 24 h dietary recall interview and obtained complete dietary data for the day before the interview. We defined diabetes as any of 10 identified types of antidiabetic agents prescribed for NHANES respondents: sulfonylureas, biguanides, insulin, alpha-glucosidase inhibitors, thiazolidinediones, meglitinides, dipeptidyl peptidase 4 inhibitors, amylin analogs, GLP-1 receptor agonists, and SGLT-2 inhibitors. Stroke definition was based on self-reported information regarding prior diagnosis of this condition. Cardiovascular disease was defined using self-reported information regarding prior diagnosis of coronary heart disease (CHD), myocardial infarction, or angina pectoris. Nine drugs were identified as other antihyperlipidemic agents: cholestyramine, clofibrate, gemfibrozil, probucol, colestipol, fenofibrate, and colesevelam.

### 2.5. Statistical Analysis

All statistical analyses were conducted using Stata, Version 16.0 (StataCorp, College Station, TX, USA), which made it possible to account for the complex survey design. Taylor series linearization was used for variance estimation. Depending on the included variables, we used an appropriate weight for each part of the analysis. These weights accounted for unequal probabilities of selection and non-response to establish an unbiased national estimate. We combined 10 cycles of NHANES data (from 1999–2000 to 2017–2018) to conduct trend analyses. Descriptive statistics on statin user characteristics were calculated separately for each weight category and survey cycle.

Primary outcome variables were used to estimate proportions and characteristics of patients who lacked awareness of their hypercholesterolemia status and/or medication. Using the *t*-test for continuous variables and the chi-square test for categorical variables, statin users showing this lack of awareness were compared with statin users who indicated different levels of awareness. We then conducted a multivariable logistic regression analysis to evaluate the odds ratios (ORs) of various risk factors associated with the lack of awareness. We included age, sex, race and ethnicity, educational attainment, number of people per household, economic status, number of prescription drugs taken, and diagnoses of hypertension, diabetes, and/or cardiovascular disease. We set the significance level to 0.05 for all analyses in this study.

## 3. Results

A total of 8798 participants were included in this study, with the distribution across the survey years shown in Table 1. The percentage of statin users in our sample of NHANES 1999–2018 survey participants more than doubled during the observation period, increasing from 7.4% to 17.3%, with an overall average of 14.6%. Among statin users, 17.6% lacked awareness of their hypercholesterolemia status or statin medications. In contrast with the proportion of statin users, the proportion of subjects who lacked this awareness did not show an annual increase from 1999 to 2018.

Table 2 shows the characteristics of two groups of statin users: those with and those without awareness. The group lacking awareness contained higher proportions of older people and of those diagnosed with diabetes and/or cardiovascular disease. Values for total cholesterol and LDL-C were also significantly lower in the group lacking awareness. The two groups had significant differences with regard to ethnicity, type of medical institution visited, taking multiple medications, and hypertension. However, there was no significant difference with regard to language usually spoken, educational background, number of people living in the respondent’s household, presence of health insurance, economic status, caloric intake, fat intake, or smoking status.

Table 3 shows the results of the multivariable logistic regression analysis for lack of awareness of hypercholesterolemia or statin medication, presenting the association with each variable. Among those indicating a lack of awareness, there was a high OR for the following: 20–40 years old vs. >75 years old (OR = 2.53; 95% confidence interval (CI): 1.27–5.05), female vs. male (OR = 1.59; 95% CI: 1.33–1.90), and non-Hispanic White vs. non-Hispanic Black (OR = 1.29; 95% CI: 1.08–1.53). Compared with taking one or two prescription drugs, there was an OR of 1.78 (95% CI: 1.36–2.32) for taking three or four tablets and of 1.80 (95% CI: 1.37–2.38) for taking five or more tablets. Compared with having none of the examined diagnoses, an OR of 1.59 (95% CI: 1.33–1.89) was seen for diabetes diagnosis, and an OR of 1.47 (95% CI: 1.20–1.79) was seen for cardiovascular disease diagnosis. Compared with visiting clinics or health centers, the OR for visiting other places was 1.50 (95% CI: 1.00–2.25). Compared with single-statin use, the OR for other antihyperlipidemic agent use was 0.51 (95% CI: 0.35–0.77). Among those taking statins, 54.3% had no previous diagnosis of diabetes or cardiovascular disease; among this group, 18.9% were >75 years old. Those lacking awareness, who had not been diagnosed with diabetes, CHD, or stroke but who were using statins for primary prevention purposes accounted for 34.0%; of these, 27.1% were >75 years old.

The number of statin users in 2017–2018 was estimated to be 40,975,232. Among this group, the number of subjects lacking awareness of their own hypercholesterolemia or statin medication tended to increase over the study period of 1999–2018 and was estimated to be 8,485,637 in 2017–2018 (Figure 2). Furthermore, among this group, we estimated that 2,517,934 had not been diagnosed with diabetes, CHD, or stroke, and 774,337 were >75 years old.

## 4. Discussion

Our findings suggest that one in six adults in the United States surveyed during the examined period—about 8.5 million people—lacked awareness about their hypercholesterolemia or statin use. The total number of those lacking awareness has risen along with the rising number of statin users, whereas the proportion lacking awareness has not changed significantly over the last two decades. A diagnosis of diabetes or cardiovascular disease, older age, male sex, non-Hispanic Black race/ethnicity, lower total cholesterol levels, lower LDL-C levels, routinely visiting the hospital emergency room for healthcare, and taking a larger number of prescribed drugs were positively associated with lack of awareness about hypercholesterolemia or statin medication. To our knowledge, ours is the first study based on nationally representative data to investigate the actual condition of patients lacking awareness about their hypercholesterolemia or statin use, despite their being prescribed statins.

The present study covered the time period from 1999 to 2018; during this time period, the fire-and-forget approach was proposed in 2002 [6] and entered into US guidelines for the treatment of hypercholesterolemia in 2013 [7]. We understand that most patients who lacked awareness of hypercholesterolemia or statin medication were diagnosed and started medication before the fire-and-forget approach was adopted in the guidelines; we were motivated to investigate how many patients actually “forgot” or did not recognize their disease or medication during this time period. We found that a considerable number of patients did not recognize their hypercholesterolemia or statin medication even during the study period. From 2014 to 2018, after the fire-and-forgot approach was recommended, the number of people who did not recognize their hypercholesterolemia or statin medication may have gradually increased, especially for those diagnosed with diabetes. Further studies should investigate awareness after the guideline revision in the near future.

Previous studies have found lack of medication knowledge to be associated with low adherence to medication regimens for chronic diseases [17,18]. A study on a Swiss population also found that, compared with those lacking awareness of their dyslipidemia, those who were aware of their dyslipidemia may have a healthier diet—but only if they self-reported being on a hypolipidemic diet [8]. Additionally, reduced adherence to statin use has been found to be associated with higher LDL levels [8,19,20]. Therefore, we hypothesized that patients lacking awareness of hypercholesterolemia or statin use have poorer control of hypercholesterolemia than do those who have awareness regarding their disease and medication. On the contrary, we found that people lacking awareness had significantly lower total cholesterol and lower LDL-C levels, compared with those with awareness, and there were no differences in body mass index, caloric intake, or fat intake. At least on the basis of the results of the present study, lack of awareness may not undermine adherence or lifestyle, but we need to be careful about the possibility of confounding or reverse causality.

Several of our findings on risk factors are compatible with previous reports. Awwad et al. [17] reported that younger age, higher education levels, and taking fewer medications were predictors of a patient’s higher level of medication knowledge. However, Hope et al. [21] reported that there was no relationship between polypharmacy and statin adherence. In the present study, lack of awareness of one’s own hypercholesterolemia or statin use was positively correlated with older age and with taking a larger number of prescribed drugs. Likewise, in the ESC/EAS Guidelines for the management of dyslipidemia [22], older age and complex polypharmacy are listed as predictors of statin non-adherence.

Additionally, the present study estimated that around 600,000 statin users >75 years old in the United States who have not been diagnosed with diabetes, CHD, or stroke lack awareness of their own hypercholesterolemia and/or statin use. There is limited evidence supporting the efficacy of statins for primary prevention of ASCVD among patients aged ≥65 years [23]. There is concern about the risk of statin-related adverse effects in older adults [24]. Myalgias, a major reason for discontinuation of statin use, are associated with interactions with many commonly used drugs [25], which may increase adverse effects in older patients [24]. Additionally, a study simulating the effect of statins for the primary prevention of ASCVD in adults aged 75–94 years suggested that statins, compared with non-statin therapies, are controversial with regard to cost-effectiveness in this population [26]. In addition to these increased adverse effects and lower cost-effectiveness, patients aged >75 years with hypercholesterolemia who have not been diagnosed with diabetes or cardiovascular disease may not need to take statins, especially when they do not understand their disease or medication. However, it might be reasonable for a patient with a relatively high risk to continue statin use even without awareness of the underlying disease or medication. Adverse events, medical economics, and ethical issues need to be considered when deciding the appropriateness of medication for older adult patients. It would be most important to show that so many people did not recognize their hypercholesterolemia or statin use. Additionally, many of such patients were not recommended in the guidelines. Therefore, healthcare providers should first inform such patients that they are taking the medication and its purpose, then confirm its current indication and discuss whether they should continue taking the drug.

Our study has several limitations. First, in the examined population, trained interviewers verified statin use by examining patients’ medication containers during home interviews. As we investigated the lack of patient awareness of the targeted conditions, it is possible that some patients forgot to present their statin medications to the interviewer. However, as medication containers were confirmed based on prescriptions, the presence of any undeclared medications other than those indicated by the containers also raises the possibility that statins were not taken daily. Second, as the 2013 American College of Cardiology/American Heart Association guidelines on the treatment of blood cholesterol began recommending statin prescriptions for patients with CHD, stroke, or diabetes, irrespective of cholesterol level, some patients may have started taking statins even without hypercholesterolemia. For most of our study period, however, the guidelines [27] recommended statin medication to target appropriate LDL-C levels in accordance with the degree of risk. Patients should have taken statins in relation to their own hypercholesterolemia status, and therefore, should have recognized their hypercholesterolemia status during that period. Even after the guidelines changed in 2013 to recommend statin regimens start at the time of ASCVD or diabetes diagnosis regardless of control of cholesterol levels, patients who initiated statin medication before 2013 should not have forgotten their hypercholesterolemia diagnosis because of the guideline revisions. Third, the repeated cross-sectional study design was observational and, therefore, did not permit assessment of a causal direction, which is potentially achievable with longitudinal data. Nevertheless, interesting findings with significant implications can be found from the assessment of changing perceptions and connections from different perspectives at the population level against the backdrop of changes in the background and environment of clinical settings. Fourth, we defined diabetes using the prescription of antidiabetic agents. This method, however, cannot identify patients with diabetes who are not prescribed these medications. We used self-reported data on hypertension, CHD, and stroke because there are weaker relationships between these conditions and particular prescription medications, compared with the relationships between statins and hypercholesterolemia and between antidiabetic agents and diabetes. For instance, regarding hypertension, diuretics are prescribed not only for high blood pressure but also for conditions such as heart failure and edema, and beta-blockers are also prescribed for conditions such as heart failure, tachycardia, and hyperthyroidism. Additionally, because the information on nutrients was collected through dietary recall interviews, the results are potentially subject to social desirability bias [28]. We expect to conduct further analyses using other databases in the near future. Fifth, statin users who answered “no” to the question about whether they were taking the medication prescribed for their condition could not be strictly identified as lacking awareness or having low adherence. Thus, we conducted a sensitivity analysis that excluded these patients, but the results remain the same.

## 5. Conclusions

Our study shows that, in the United States, one in six statin users, or about 8.5 million people, lacked awareness of their hypercholesterolemia status or statin use, despite taking statins. However, this lack of awareness was not associated with poor cholesterol control. Future guideline revisions should be accompanied by discussions on whether it is better to prioritize awareness of the underlying disease or to provide medication without promoting such awareness.

## Figures and Tables

**Figure 1 ijerph-19-06099-f001:**
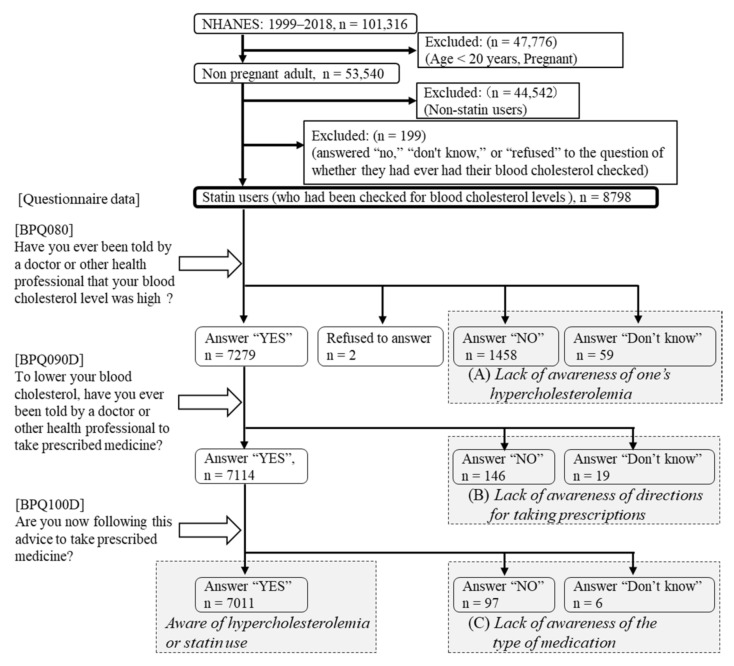
Study flow diagram. Abbreviation: NHANES, National Health and Nutrition Examination Survey. Statin users were classified in accordance with their responses. Lack of awareness of one’s own hypercholesterolemia or statin use was regarded as the sum of (**A**) lack of awareness of one’s hypercholesterolemia; (**B**) lack of awareness of directions for taking their prescription; and (**C**) lack of awareness of the type of medication.

**Figure 2 ijerph-19-06099-f002:**
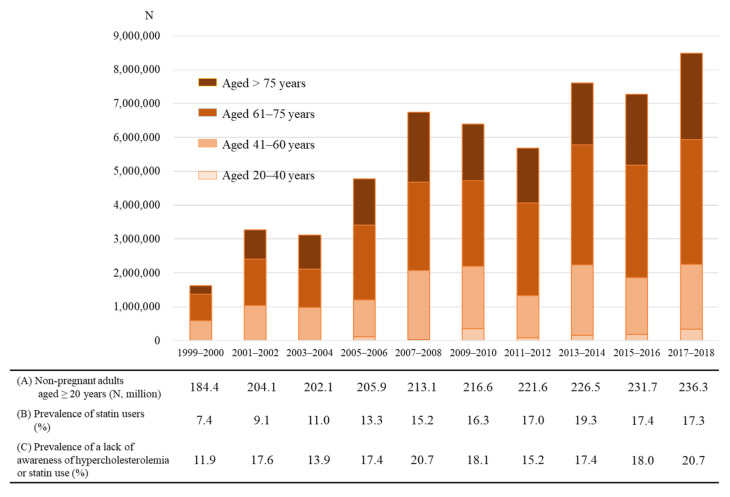
Trends among statin users lacking awareness of their hypercholesterolemia or statin use. The number of statin users lacking awareness of their hypercholesterolemia or statin use was estimated by multiplying (**A**) non-pregnant adults, (**B**) the number of statin users, and (**C**) the number lacking awareness of their hypercholesterolemia or statin use.

**Table 1 ijerph-19-06099-t001:** Characteristics of the study sample by survey cycle for 1999–2018.

Characteristic	1999–2000	2001–2002	2003–2004	2005–2006	2007–2008	2009–2010	2011–2012	2013–2014	2015–2016	2017–2018	Total
Unweighted sample no.	367	529	646	696	1072	1141	993	1123	1061	1170	8798
Weighted proportion of statin uses among non-pregnant adults	7.4	9.1	11.0	13.3	15.2	16.3	17.0	19.3	17.4	17.3	14.6
Age range, y ^a^											
20–40	1.3	4.9	3.3	3.9	4.1	4.0	2.9	3.7	2.1	3.4	3.4
41–60	44.2	42.5	40.4	35.5	35.4	38.3	39.4	35.0	29.7	28.5	35.7
61–75	44.0	35.7	40.2	40.3	40.2	39.5	40.1	42.6	47.9	45.1	42.0
>75	10.5	16.9	16.2	20.3	20.2	18.2	17.6	18.8	20.3	23.0	19.0
Male sex	51.9	54.9	52.5	51.2	51.8	54.7	49.2	52.6	56.4	53.3	52.9
Race/ethnicity ^a^											
Non-Hispanic White	83.9	86.3	83.5	80.4	80.4	78.1	75.3	77.8	73.1	70.0	77.6
Non-Hispanic Black	4.8	5.9	7.3	9.6	8.5	9.2	10.6	10.0	9.5	9.3	9.0
Mexican American	2.0	2.1	3.0	2.7	3.9	4.9	3.0	4.0	5.1	5.0	3.9
Others ^b^	9.3	5.7	6.1	7.2	7.2	7.8	11.1	8.2	12.2	15.7	9.6
Language usually spoken ^a^											
English	94.9	96.1	95.6	96.1	93.8	91.5	92.2	94.2	90.9	89.6	93.5
Spanish	3.6	1.8	1.9	1.3	3.7	4.9	4.8	2.3	4.8	4.8	3.4
Others	1.4	2.1	2.5	2.6	2.5	3.7	3.0	3.5	4.3	5.5	3.1
Educational attainment ^a^											
>High school	42.0	53.3	45.4	50.5	49.5	55.4	58.4	62.0	62.0	57.8	55.4
High school or GED	31.5	27.2	33.6	30.9	29.8	24.0	22.1	22.0	23.1	29.9	26.6
<High school	26.5	19.5	21.0	18.6	20.7	20.6	19.5	16.0	15.0	12.3	18.1
Number of people per household ^a^											
1	17.3	18.9	21.5	22.0	16.8	18.8	17.0	18.5	21.7	18.2	19.0
2	55.1	49.0	51.6	49.6	52.7	49.7	48.3	50.1	49.0	48.9	50.0
≥3	27.6	32.1	26.9	28.4	30.5	31.5	34.7	31.4	29.3	32.9	30.9
Family income category ^a^											
4 of IPR or greater	34.0	48.8	36.2	37.0	40.2	42.4	42.1	38.7	39.1	41.7	40.2
2 to <4 of IPR	37.4	24.1	33.2	34.0	28.8	30.5	26.3	29.8	28.3	29.3	29.7
1 to <2 of IPR	18.9	20.2	22.4	22.0	21.9	18.5	17.8	20.3	21.6	20.2	20.3
<1 of IPR	9.7	6.9	8.2	7.0	9.1	8.6	13.8	11.2	11.1	8.9	9.8
Health insurance	97.7	98.0	95.1	95.8	95.0	93.9	95.4	96.5	97.4	97.1	96.1
Routine place to go for healthcare ^a^											
Clinic or health center	13.8	20.6	13.5	15.3	13.5	17.7	17.9	9.5	26.0	16.6	16.6
Doctor’s office or HMO	79.7	75.9	79.8	79.6	82.6	78.7	77.2	87.0	65.7	76.4	78.2
Other places	4.8	3.1	4.5	4.2	1.9	2.2	3.3	1.7	5.5	4.4	3.5
No place	1.7	0.4	2.1	0.9	2.0	1.3	1.6	1.8	2.8	2.6	1.8
Number of prescription drugs ^a^											
1–2	26.8	19.8	19.8	19.6	20.4	20.1	20.7	16.5	15.4	15.6	18.7
3–4	30.7	32.0	28.7	32.3	29.0	33.0	29.5	31.0	30.1	26.9	30.2
≥5	42.5	48.2	51.5	48.2	50.6	46.9	49.8	52.5	54.5	57.5	51.1
BMI, mean (SD), kg/m^2^	29.2 (4.6)	29.5 (5.4)	29.7 (5.3)	30.5 (5.5)	30.4 (6.4)	30.6 (6.2)	29.9 (5.5)	30.6 (5.8)	30.9 (6.0)	30.8 (5.8)	30.4 (5.8)
Total cholesterol level, mean (SD), mg/dL	201.8(32.6)	196.2(36.3)	191.1(38.3)	183.2(35.7)	177.1(36.9)	178.0(34.4)	180.6(33.4)	174.4(33.6)	172.6 (31.0)	169.7 (35.4)	179.4(35.8)
LDL-C level, mean (SD), mg/dL	119.4(31.2)	113.1(26.5)	100.2(27.0)	97.0(29.6)	95.8(30.2)	98.8(27.9)	100.7(27.7)	95.1(24.8)	93.5 (23.6)	90.7 (28.5)	98.0(28.2)
Caloric intake, mean (SD), kcal/d	1857.8(660.7)	1915.5(796.5)	1945.9(748.7)	1977.9(719.9)	1927.9(807.8)	2003.0(807.5)	1954.4(671.7)	1974.6(755.9)	2008.6 (712.5)	2063.9 (792.9)	1976.1(754.2)
Fat intake, mean (SD), g/d	68.5 (31.5)	72.7 (36.5)	75.5 (37.2)	76.8 (33.7)	76.7 (41.8)	77.6 (40.9)	74.8 (34.8)	80.2 (40.1)	81.7 (36.2)	85.8 (38.8)	78.2 (37.9)
Lack of awareness of one’s hypercholesterolemia or statin use	11.9	17.6	14.0	17.4	20.8	18.2	15.1	17.4	18.0	20.7	17.6
(Lack of awareness of one’s hypercholesterolemia)	(9.1)	(14.1)	(10.7)	(15.0)	(18.0)	(16.5)	(13.2)	(14.7)	(15.3)	(18.1)	(15.0)
(Lack of awareness of directions for taking prescription)	(1.7)	(1.6)	(1.3)	(1.9)	(1.6)	(0.9)	(1.2)	(1.4)	(1.5)	(2.1)	(1.5)
(Lack of awareness of medication)	(1.1)	(1.9)	(1.9)	(0.5)	(1.1)	(0.7)	(0.8)	(1.3)	(1.3)	(0.5)	(1.0)
Hypertension diagnosis	59.8	58.0	67.5	63.7	63.2	66.8	66.0	68.1	68.8	67.1	65.8
Diabetes diagnosis	21.3	23.6	27.6	29.2	29.5	29.4	30.8	32.8	38.5	39.8	31.7
Cardiovascular disease diagnosis	37.0	36.3	31.6	31.0	27.2	25.3	27.4	26.6	27.8	31.3	29.2
Other antihyperlipidemic agents use	3.4	3.5	6.7	7.1	8.3	7.9	5.8	8.6	6.6	4.8	6.6
Smoking ^a^											
Never smoked	41.2	42.2	42.9	42.0	47.3	49.4	47.1	49.0	44.8	51.5	46.6
Former smoker	42.9	42.6	41.6	42.6	39.7	38.1	36.6	36.3	40.3	35.7	38.9
Current smoker	15.9	15.2	15.5	15.4	13.0	12.6	16.3	14.8	15.0	12.9	14.5

Abbreviations: HMO, health maintenance organization; IPR, income-to-poverty ratio; BMI, body mass index; LDL, low-density lipoprotein. ^a^ Percentages do not total 100% because of rounding. ^b^ This category includes other Hispanics and other races, including multiracial participants.

**Table 2 ijerph-19-06099-t002:** Characteristics of the study sample by awareness of one’s hypercholesterolemia status or statin use.

Characteristic	(1) Aware of Hypercholesterolemia and Statin Use	(2) Lack of Awareness of One’s Hypercholesterolemia or Statin Use	*p*-Value ^a^	(2)/(1) + (2)
Unweighted sample no.	7011	1785		-
Weighted proportion among statin users	82.4	17.6		-
Year ^b^				
1999–2000	4.7	2.9	0.11	11.9
2001–2002	5.9	5.9		17.6
2003–2004	7.5	5.6		14.0
2005–2006	8.8	8.7		17.4
2007–2008	10.0	12.3		20.8
2009–2010	11.2	11.6		18.1
2011–2012	12.4	10.3		15.1
2013–2018	14.1	13.8		17.4
2015–2016	12.8	13.2		18.0
2017–2018	12.6	15.4		20.7
Age range, y ^b^				
20–40	3.6	2.3	<0.001	12.3
41–60	37.7	26.3		13.0
61–75	41.7	43.5		18.2
>75	17.1	27.8		25.9
Male	51.1	61.2	<0.001	17.6
Race/ethnicity ^b^				
Non-Hispanic White	78.3	74.6	<0.001	16.9
Non-Hispanic Black	8.7	10.1		19.8
Mexican American	3.9	3.9		17.7
Others ^c^	9.2	11.5		21.1
Language usually spoken ^b^				
English	93.8	91.8	0.08	17.8
Spanish	3.6	3.9		18.9
Others	3.2	4.3		22.6
Educational attainment ^b^				
>High school	56.0	52.7	0.07	16.8
High school or GED	26.5	26.8		17.7
<High school	17.5	20.5		20.0
Number of people per household ^b^				
1	19.1	18.6	0.91	17.2
2	50.0	50.4		17.7
≥3	30.9	31.0		17.6
Family income category ^b.^				
4 of IPR or greater	40.7	38.0	0.14	16.5
2 to <4 of IPR	30.0	28.6		16.8
1 to <2 of IPR	19.9	22.2		19.1
<1 of IPR	9.5	11.1		19.7
Health insurance	96.1	96.3	0.75	17.6
Routine place to go for healthcare ^b^				
Clinic or health center	16.8	15.3	0.006	16.3
Doctor’s office or HMO	78.1	78.2		17.6
Other places	3.1	5.1		25.8
No place	1.9	1.4		13.2
Number of prescription drugs ^b^				
1–2	20.7	12.0	<0.001	11.2
3–4	30.0	30.7		17.9
≥5	49.8	57.3		19.7
BMI, mean (SD), kg/m^2^	30.4 (5.7)	30.3 (6.6)	0.88	-
Total cholesterol level, mean (SD), mg/dL	182.7 (34.9)	164.1 (36.7)	<0.001	-
LDL-C level, mean (SD), mg/dL	100.7 (27.6)	85.5 (28.0)	<0.001	-
Caloric intake, mean (SD), kcal/d	1962.6 (738.2)	1923.3 (818.9)	0.26	-
Fat intake, mean (SD), g/d	77.4 (37.1)	76.4 (42.8)	0.58	-
Hypertension diagnosis	66.5	62.3	0.03	17.5
Diabetes diagnosis	29.7	41.5	<0.001	23.0
Cardiovascular disease diagnosis	26.9	39.5	<0.001	17.6
Other antihyperlipidemic agents use	7.1	4.3	0.006	11.6
Smoking ^b^				
Never smoked	46.7	46.3	0.98	17.5
Former smoker	38.9	39.1		17.7
Current smoker	14.5	14.5		17.7

Abbreviations: HMO, health maintenance organization; IPR, income-to-poverty ratio; BMI, body mass index; LDL, low-density lipoprotein. ^a^ The chi-squared test was used for categorical variables. ^b^ Percentages do not total 100% because of rounding. ^c^ This category includes other Hispanics and other races, including multiracial participants.

**Table 3 ijerph-19-06099-t003:** Odds ratios for lack of awareness of one’s hypercholesterolemia status or statin use.

	Outcome Variable: Lack of Awareness of One’s Hypercholesterolemia or Statin Use		
Independent Variable	OR	(95% CI)	*p*-Value ^a^
Year			
1999–2000	1	(ref)	
2001–2002	1.36	(0.76–2.43)	0.31
2003–2004	1.05	(0.56–1.98)	0.88
2005–2006	1.30	(0.72–2.32)	0.38
2007–2008	1.54	(0.88–2.72)	0.13
2009–2010	1.46	(0.84–2.53)	0.18
2011–2012	1.08	(0.60–1.93)	0.80
2013–2018	1.24	(0.69–2.26)	0.47
2015–2016	1.18	(0.63–2.20)	0.60
2017–2018	1.49	(0.83–2.68)	0.19
Age range, y ^b^			
20–40	1	(ref)	
41–60	1.13	(0.60–2.17)	0.70
61–75	1.64	(0.83–3.24)	0.15
>75	2.53	(1.27–5.05)	0.009
Male sex (vs. female sex)	1.59	(1.33–1.90)	<0.001
Race/ethnicity ^b^			
Non-Hispanic White	1	(ref)	
Non-Hispanic Black	1.29	(1.08–1.53)	0.006
Mexican American	1.12	(0.80–1.56)	0.52
Others ^c^	1.40	(1.08–1.83)	0.01
Language usually spoken ^b^			
English	1	(ref)	
Spanish	0.98	(0.70–1.36)	0.88
Others	1.10	(0.72–1.66)	0.67
Educational attainment ^b^			
>High school	1	(ref)	
High school or GED	1.11	(0.90–1.36)	0.34
<High school	1.05	(0.85–1.30)	0.65
Family income category ^b^			
4 of IPR or greater	1	(ref)	
2 to <4 of IPR	0.86	(0.70–1.07)	0.18
1 to <2 of IPR	0.95	(0.76–1.18)	0.62
<1 of IPR	1.09	(0.82–1.44)	0.58
Routine place to go for healthcare ^b^			
Clinic or health center	1	(ref)	
Doctor’s office or HMO	1.13	(0.92–1.39)	0.23
Other places	1.50	(1.00–2.25)	0.05
No place	0.84	(0.48–1.44)	0.51
Number of prescription drugs ^b^			
1–2	1	(ref)	
3–4	1.78	(1.36–2.31)	<0.001
≥ 5	1.80	(1.37–2.38)	<0.001
Hypertension diagnosis (vs. no diagnosis)	0.63	(0.52–0.77)	<0.001
Diabetes diagnosis (vs. no diagnosis)	1.59	(1.33–1.89)	<0.001
Cardiovascular disease diagnosis (vs. no diagnosis)	1.47	(1.20–1.79)	<0.001
Other antihyperlipidemic agents use (vs. single-statin use)	0.51	(0.35–0.77)	0.01

Abbreviations: OR, odds ratio; CI, confidence interval; IPR, income-to-poverty ratio. ^a^ Adjusted for year, age, sex, race, language usually spoken, educational attainment, family income, number of prescription drugs, hypertension diagnosis, diabetes diagnosis, and cardiovascular disease diagnosis. ^b^ Percentages do not total 100% because of rounding. ^c^ This category includes other Hispanics and other races, including multiracial participants.

## Data Availability

The NHANES survey data are available publicly. https://www.cdc.gov/nchs/nhanes/index.htm (accessed on 13 May 2022).

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
