# Peer review of "Lack of Awareness of Own Hypercholesterolemia or Statin Medication among Adult Statin Users in the United States: Prevalence and Patient Characteristics in a Repeated Cross-Sectional Study"

_ijerph, 2022, doi:10.3390/ijerph19106099_

Round 1

Reviewer 1 Report

The authors analysed the awareness of patients' own hypercholesterolemia or statin treatment among statin users. The topic is relevent and interesting. The strength of the study is the number of interviewed patients. I have only some minor concerns.

  1. I would suggest to add a sentence regarding the approach to cholesterol treatment in the latest ACC/AHA cholesterol guidelines in the introduction. The authors mentioned 2002 and 2013 guidelines. Has the approach been changed since 2013?
  2. The authors did not describe if patients on statins were treated with other cholesterol-lowering drugs. It would be an interesting information to add.
  3. In the line 269 "people who did not recognize these things" do you mean the lack of awareness of their own hypercholesterolemia and statin medication? In my opinion this sentence is not clear for the reader.
  4. Could the authors suggest other solutions to this problem apart from guidelines revision? What can general practioners do to help to ameliorate the situation? The lack of patients' awareness regarding statin medication is alarming and appropriate measures should be implemented.

Author Response

We appreciate your thoughtful comments. Please see the attachment.

Reviewer 2 Report

The paper is straightforward and simple, but nonetheless relevant given that dyslipidemia is prevalent. This paper is acceptable for publication but these are some of the comments and recommendations for this paper. 

  1. General writing style can be improved, examples given below
    • On page 4, “Covariates”- the words, “We categorized “ was used 3x to start sentences in this paragraph, when in fact the ideas can be conveyed better. On the next paragraph, the same words appear again, and also “We defined... (2x)”
  2. Abstract
    • Typically for original researches, most journals would prefer a structured abstract. Would recommend that the authors make a structured abstract with the following main sections: Aims/Background, Methods (or methodology), Results and Conclusions.
    • The last sentence of the abstract may need to be supported by results:

       “ These patients’ statin use may be controversial because of the risks of statins, especially for those with lower cardiovascular disease risk.”  The authors need to first mention the proportion of subjects who were of low CV risk and then mention why giving statins is controversial for this subset of patients. Or rewrite it to refer to the population aged 75 years and above.

  • There are 7 keywords. We typically recommend only 4-5 key terms.
  1. Introduction: adequate and substantial
  2. Materials and methods
    • Rewrite line 90 as the idea of the sentence is not clear. “A final sample of 8,798 statin users resulted. “ Maybe say, “We included a final sample of 8,798 statin users for this study.” However, this can also be in the results rather than in the materials and methods section.
    • The definitions and categories of various outcomes and co-variates are well appreciated. However, based on the abstract, CV risk categories was also considered as one of the co-variates of statin use. How was this categorized or defined into risk categories?
  3. Results
    • Typically, the results should usually first show the general characteristics of the study subjects especially as derived from Table 1. So, this section could have started by introducing Table 1. Something like, “A total of 8798 participants were included in this study with the distribution across the survey years shown in Table 1.
    • Since the survey year was actually not included as a factor in Table 2, why was it then analyzed in the multivariate logistic regression as a possible factor associated with lack of awareness of statin use or dyslipidemia diagnosis?
  4. Discussion
    • The discussion is generally comprehensive and more than adequate.

Author Response

(The authors gave the same response as above.)
